# Heat Shock Protein Response to Stress in Poultry: A Review

**DOI:** 10.3390/ani13020317

**Published:** 2023-01-16

**Authors:** Krishnan Nair Balakrishnan, Suriya Kumari Ramiah, Idrus Zulkifli

**Affiliations:** 1Laboratory of Sustainable Animal Production and Biodiversity, Institute of Tropical Agriculture and Food Security, Universiti Putra Malaysia (UPM), Serdang 43400, Malaysia; 2Department of Animal Science, Faculty of Agriculture, Universiti Putra Malaysia (UPM), Serdang 43400, Malaysia

**Keywords:** poultry, heat shock proteins, thermal stress, non-thermal stress, poultry well-being

## Abstract

**Simple Summary:**

The demand for poultry production is growing along with the human population, and supplying sufficient animal protein remains an ultimate priority. Securing food for future generations may have wide-ranging implications for poultry well-being. Heat shock proteins (HSPs) have been studied extensively in poultry to assess stress levels in different circumstances. However, the information on HSP expression is scattered; therefore, the present review attempts to signify the importance of HSPs in various conditions, focusing mainly on poultry. Considering the importance of poultry well-being, the synthesis and release of HSPs are primarily associated with the birds’ ability to cope with different stress conditions.

**Abstract:**

Compared to other animal species, production has dramatically increased in the poultry sector. However, in intensive production systems, poultry are subjected to stress conditions that may compromise their well-being. Much like other living organisms, poultry respond to various stressors by synthesising a group of evolutionarily conserved polypeptides named heat shock proteins (HSPs) to maintain homeostasis. These proteins, as chaperones, play a pivotal role in protecting animals against stress by re-establishing normal protein conformation and, thus, cellular homeostasis. In the last few decades, many advances have been made in ascertaining the HSP response to thermal and non-thermal stressors in poultry. The present review focuses on what is currently known about the HSP response to thermal and non-thermal stressors in poultry and discusses the factors that modulate its induction and regulatory mechanisms. The development of practical strategies to alleviate the detrimental effects of environmental stresses on poultry will benefit from detailed studies that describe the mechanisms of stress resilience and enhance our understanding of the nature of heat shock signalling proteins and gene expression.

## 1. Introduction

The poultry industry is a growing part of global agribusiness. The production level reflects high biological and economic performance, contributing to cheap and abundant food and improved quality of life. World poultry production soared from 9 to 133 million tonnes between 1961 and 2020, and egg production increased from 15 to 93 million tonnes [1]. Compared to other farm animal species, production has dramatically increased in the poultry sector [2]. However, in general, the environment of intensively raised poultry is a composite of interacting stressors that, in the broadest sense, can include all conditions in which the bird lives, including external (e.g., temperature, light, social conditions; human–animal interactions), as well as internal conditions (e.g., pathogens; toxins). Poultry try to cope with these highly demanding conditions (stressors) using behavioural and physiological stress responses to restore homeostasis. Therefore, a bird’s success in adapting to its environment depends on its severity and the physiological ability of the bird to respond appropriately. All species must have the appropriate physiological reactions to environmental and homeostatic stresses to survive and thrive.

According to Broom [3], “stress is an environmental effect on an individual which overtaxes its control systems and results in adverse consequences, eventually reduced fitness.” Koolhas et al. [4] suggested that “the term ‘stress’ should be restricted to conditions where an environmental demand exceeds the natural regulatory capacity of an organism, in particular situations that include unpredictability and uncontrollability.” Coordinating and managing the neuroendocrine and autonomic stress systems are necessary for restoring and maintaining homeostasis [5]. The autonomic nervous system (ANS) and the hypothalamic–pituitary–adrenal (HPA) axis influence physiological stress responses in mammalian and avian species. The ANS system releases epinephrine (adrenaline) from the medulla or centre of the adrenal gland, as part of the sympathetic division of the autonomic nervous system [6]. The rapid mobilisation of metabolic resources and the coordination of the fight/flight response are made possible by increased circulating adrenaline. On the other hand, the HPA system produces glucocorticoids, steroid hormones that include corticosterone (CORT) in avian species.

A generalised stress response occurs at the cellular level to maintain homeostasis. The cellular stress reaction depends on the integrity and role of proteins. The production of heat shock proteins (HSPs) in response to stressors that threaten the cell’s life is one of the most common characteristics of the cellular stress response. Heat shock proteins profoundly modify the physiological stress response and encourage stress tolerance [7,8]. Heat shock or stress-induced proteins are a group of phylogenetically conserved proteins distributed among diverse cellular organisms. Because HSPs have been linked to a critical role in cellular resistance to physical and chemical stresses, studying these proteins in living organisms is crucial. It has been proposed that the levels of HSP production or accumulation can help to identify if an organism finds a specific environment stressful. Figure 1 is a generalised illustration of the activation of genes that encode HSP expression in response to stress. During stress, the HSP–heat shock factor (HSF) complex dissociates to reach an active state in the cytosol. The HSF–phosphorylated trimer complex enters the nucleus and activates the HSP gene by binding to the promoter’s heat shock element (HSE). This leads to the proliferation of HSPs in the cell to help refold the damaged proteins.

Furthermore, the close relationship between the accumulation of HSPs and the organism’s physiological state suggests that the HSP response can be more sensitive than the existing stress indicators. Heat shock proteins are activated in response to thermal and non-thermal stressors [9,10,11]. Ritossa [12] laid the foundation for discovering HSPs by observing the chromosomal puffs or focal swellings in the salivary gland of *Drosophila busckii* soon after the induction of temperature shock. The genes responsible for these swellings were first identified as “heat shock loci” and were believed to be unique to fruit flies. Later, Tissieres et al. [13] described the sites of the transcriptional induction of genes that encode a group of particular proteins known as HSPs. Gradually, HSPs were identified in every organism, from archaebacteria to animals, with specific responses that vary between organisms [14]. HSPs are classified systematically based on molecular weight. HSPs with a large molecular weight include families HSP110, HSP90, HSP70, and HSP60, while small HSPs include HSPB1–HSPB10 with a molecular weight of 8–30 kDa. In addition, glucose-regulated proteins were identified as minor HSPs with a molecular weight of 34, 47, 56, 75, 78, 94, and 174 kDa [15]. In chickens, the *HSP70* gene is located on the fifth chromosome and has a 1905-bp coding region length with only one exon [16].

However, polymorphisms are frequently encountered in this gene region, resulting in different HSP70 genotypes [17]. This phenomenon contributes to the ability of chickens to tolerate stressors differently. Generally, HSPs have proven their protective role in various cellular processes, ensuring cells’ survival during stress.

Moreover, HSPs are well-known to participate in protein secretion, assembly, maintaining the integrity of structural proteins, folding, trafficking, protein degradation, and regulating transcription factors [18]. In this sense, as a molecular chaperone, HSPs are responsible for maintaining cell homeostasis by preventing cell apoptosis and promoting cell survival [19,20]. To achieve this, HSPs are actively involved in cytoprotective mechanisms, restricting protein aggregation and preventing protein misfolding [21]. Heat shock proteins are also involved in the refolding of stress-denaturing proteins in a cell. In general, HSPs reside at the frontier of cellular defence, and become overexpressed and overproduced under stress, assuming cytoprotective roles to maintain cellular integrity and gain stress tolerance in cells, when exposed to stress for a certain period of time [20].

Over the past 60 years, poultry breeding programmes have focused on the genetic improvement of production traits, such as growth, feed efficiency, and laying rate, to meet the ever-growing global demand for animal protein [22]. However, the impact of long-term selection for high production efficiency has been linked to unfavourable consequences for poultry well-being [23]. Hence, a key issue is how poultry can adapt to the challenging environment of modern production systems. This review aims to survey what is known about HSP response to thermal and non-thermal stressors in poultry, and discuss the factors that modulate its induction and regulatory mechanisms. There is considerable evidence that suggests that, other than heat, chemicals, heavy metals, microorganism infection, cold stress, feed restriction, transportation, social isolation, higher stocking densities, and other stress-evoking factors lead to HSP production in poultry (Table 1).

## 2. Heat Stress (HS)

Dealing with the rise in global temperatures has been a major challenge for the poultry industry. Global warming increases the ambient temperature, which in turn makes heat waves more likely. On 27 May 2021, the World Meteorological Organization (WMO) updated the possibility of the average global temperature surpassing 1.5 °C above pre-industrial temperatures [76]. This phenomenon has raised worldwide concern about animal well-being, especially in the broiler chicken industry, due to the potentially adverse thermoregulatory effects of rapid growth rates and body mass increases. A considerable number of studies have been published on the adverse effects of HS on poultry, encompassing health, welfare and economic traits, and productive and reproductive performance [77,78,79]. However, the mechanisms that underpin HSP expression in response to HS are not fully understood. They vary depending on the birds’ age, gender, genotype, nutritional status, and rearing system. Therefore, this paper attempts to summarise and critically examine the findings on HSP expression.

Almost 30 years ago, a pioneering study on HSP70 and HSP90 in chickens was initiated by Kelley and Schlesinger [80], using chicken embryo fibroblast cells (in vitro) incubated at 45 °C to compare the similar response exhibited by *Drosophila*. Soon after, a few studies suggested that treatment with amino acid analogues could induce the production of three proteins with sizes of approximately 23 kDa, 70 kDa, and 90 kDa [81,82]. Subsequently, more studies were carried out to confirm the organisation, nucleotide sequence, and protein size of HSP70 [16,80,81,83]. Wang’s [84] in vivo studies on the HSP response to HS and its association with thermotolerance have created vast interest in HSP research in poultry. Wang and Edens [24] confirmed the detection of HSP70 mRNA and proteins in the testes and bursa of Fabricius in chickens. Einat et al. [25] showed that exposing broilers to acute HS for 4 h resulted in the rapid synthesis of HSP 90, 70, and 27 in the heart, muscle and lungs. Importantly, novel HSP 29 was identified after only 3 h of acute HS exposure, while Edens et al. [85] failed to detect it initially. Interestingly, both studies followed a similar protocol of peripheral blood lymphocyte isolation; however, Edens et al. [85] applied a shorter period of HS exposure (60 min), which resulted in milder hyperthermia, resulting in the absence of HSP29. The expression of HSP70 was further confirmed in myocardial cells by Yu et al. [26]. This could be attributed to the ability of HSP70 to protect the cytoskeleton structure of the heart from severe HS. The study also revealed that the induction of HSP70 mRNA and proteins were time- and tissue-dependent, concurring with the findings of Leandro et al. [43] and Zhen et al. [86]. Al-Aqil and Zulkifli [87] reported the steady production of HSP70 under stress conditions, which could aid host cells during recovery from HS [88]. However, different expression patterns of HSP70 mRNA were observed in several studies based on different conditions. Gabriel et al. [89], Yu et al. [90], and Mahmoud et al. [91] reported that the mRNA expression peaked after 3, 2, and 1 h, respectively. These discrepancies could be attributed to age, strain, and rearing system variables. Siddiqui et al. [34] investigated the expression pattern of HSPs following different exposure durations of chronic HS in the small intestine of chickens.

This study provides in-depth findings on HSP gene expression and protein production, adding more value to the concept of HSPs being time- and tissue-dependent. In detail, the expression levels of HSP70, HSP60, and HSP47 were significantly higher in the 3 h treatment group in the duodenum, jejunum, and ileum. In contrast, higher protein production of HSP70 and HSP60 was found only in the duodenum and jejunum at 6 h. This shows that HSP70 and HSP60 were enhanced during an early acute HS stage and decreased after reaching a peak [92]. On the other hand, HSP47 was significantly higher at 3 h in the duodenum and ileum. The similar expression of HSP47 was observed by Tang et al. [93], reporting that higher expression was observed during short-term HS and reduced when the exposure exceeded 5 h. The discrepancies between mRNA expression and protein expression can be explained by mRNA turnover, in which both mechanisms are unrelated [94]. Another study on laying broiler breeder chickens conducted by Xie et al. [27] demonstrated that the duration and severity of HS significantly affected the expression pattern and target tissues of HSP, particularly HSP70 and HSP90. In muscle, the mRNA expression of HSP70 and HSP90 tended to increase during acute HS, but not during chronic HS. In the liver, HSP70 and HSP90 were upregulated during acute and chronic HS, respectively. Interestingly, an opposite trend was observed for the heart, where HSP70 and HSP90 expression were upregulated during chronic and acute HS, respectively. The evidence reviewed here suggests that the muscle HSP response is less sensitive to HS compared to the heart [26,27,28,95,96]. These findings suggest that the reduced activity of muscle superoxide dismutase (SOD) may contribute to tissue damage. In addition, heat stress may adversely affect roosters’ semen quality and subsequent reproductive performance [97,98]. High temperatures might decrease the fertility of birds, and the microarray analysis showed the upregulation of HSP genes (*HSP70*, *HSP90AA1*, and *HSP25*) in heat-stressed chicken testes [29]. Similarly, Wang et al. [30] showed the induction of HSP gene expression (*HSP70*, *HSP25*, *HSP90AA1*, *HSPA8*, *HSPA5*, *HSPH1*, and *HSPD1*) in the testes of broiler-type Taiwan country chickens after 4 h of acute HS.

A recent study by Greene et al. [31] generated completely different results than those reported in previous work on heat-stressed broilers. The HSPs (90, 70, 60, and 27) and HSFs (1, 2, 3, and 4) were not significantly expressed during HS. To the authors’ knowledge, this was the only study on chickens that showed non-significant expression of any HSPs or HSFs during HS. A possible reason for the conflicting findings is that pre-acclimatised birds were used in those studies. Thus, they acquired antioxidant capacities, which helps them to manage reactive oxygen species (ROS) during HS. The second reason for the inconsistent results could be the type of cells chosen for the study. In this case, the authors selected whole blood to analyse the mRNA expression, which may not reflect the actual HSPs’ response to HS. The cell types measured are crucial, and further study should involve two or more types of cells to understand the key players in HS response [99].

Thermal manipulation during mid-stage (ED 10–18) or later-stage (ED 16–18) embryogenesis [36,100,101,102] could enhance the capacity of chickens to express HSP70 and lead to concomitant heat tolerance later in life. Al-Zghoul et al. [36] demonstrated a significant increase in HSP70 mRNA expression in post-hatched chicks’ muscles, hearts, and brains after thermal manipulation. The chicks’ muscles demonstrated the highest expression of HSP70, while the lowest was noted in brain tissue. The lower expression of HSP70 mRNA in the brain could be associated with the ability of the brain to control the core temperature (2–3 °C lower than the rectal temperature) during hyperthermia [103]. The acquisition of heat tolerance in chicks has been reviewed by Vinoth et al. [104]. The authors suggested that the promoter of the HSP70 gene was affected due to DNA methylation, which altered the protein expression and response to HS during the chicks’ post-hatch life.

Kang and Shim [32] investigated the expression of different HSPs and HSFs resulting from early heat conditioning and acute HS in broilers and showed various mRNA and protein expression patterns. Pre-heat-conditioned broilers subjected to HS demonstrated higher protein expression of HSP40, while no significant changes were noticed for HSP70 and 60. On the contrary, all HSP mRNA expressions (70, 60, 40, HSF1, and HSF3) were higher than in the non-heat-conditioned broilers and controls. In comparison, only HSP40 and HSP70 protein and mRNA expression increased in the non-heat-conditioned chickens subjected to HS. These findings are similar to those reported by Liew et al. [105] and Yahav et al. [106], which demonstrated that initial heat exposure did not significantly influence the HSP70 expression. Toplu et al. [35] also found a lower expression of HSP70 in the kidneys, brain, and liver of early heat-treated chicks after exposure to chronic HS, which is in agreement with the findings reported by Vinoth et al. [33]. It has been suggested that higher HSP expression is associated with better heat tolerance [107,108]. Prenatal HS-induced HSP70 expression in embryos [109] may enhance thermotolerance during the later stage of life in chickens. For example, Vinoth et al. [33] discovered significantly higher mRNA and protein expression of HSP90α, 90β, 70, and 60 in thermoconditioned embryo liver samples. However, chickens subjected to 42 d of chronic HS and thermal conditioning showed a significant reduction in mRNA and protein expression of HSP90β, 70, and 27 compared to the control. The group’s reduced expression of HSPs later in life showed the benefit of thermal conditioning in response to HS later in life.

## 3. Cold Stress

As is the case with HS, cold stress (CS) may also adversely affect poultry’s well-being and productivity [110]. CS may trigger immune and physiological responses, while promoting necrotic enteritis in broiler chicks [38,39,111,112,113]. Huff et al. [114] and Zhao et al. [39] reported that CS may increase birds’ susceptibility to infectious agents and elicit an inflammatory response. However, the complex molecular regulatory systems that control CS are not yet fully understood in poultry. Due to varying expression patterns, several HSPs respond to CS in various ways. Zhao et al. [38] reported that CS treatment (12 ± 1 °C) augmented the expression levels of HSPs (HSP 70, 60, 40, and 27) due to the damage caused to the antioxidant defence system in the hearts of chickens. On the contrary, the authors noted a significant decrease in HSP 90 mRNA expression following CS. Meanwhile, in immune system organs, all the mRNA expressions of HSPs (HSPs 90, 70, 60, 40, and 27) were found to be upregulated in similar CS treatments [113]. Similarly, Zhao et al. [38,113] showed that higher mRNA expression of HSPs (HSPs 90, 70, 60, 40, and 27) was often associated with the oxidative stress induced by CS. Wei et al. [40] noted that cold-acclimated broilers (for 34 days) had lower mRNA expression (of HSPs 90, 70, and 27) during CS. These findings were further supported by Su et al. [41], who noted a significantly lower expression of HSPs in the cold-stimulated group compared to the control after the acute CS. Cold stimulation seems to reduce the HSPs’ expression levels, thus alleviating potential harm from CS. Furthermore, prenatal thermal manipulation (39 °C and 65% RH for 18 h/day from embryonic days 10 to 18) reduced the expression of HSP70 in hepatic and splenic cells, while a decrease in HSF3 was found in the hepatic cells of broiler chicks [42]. Interestingly, for lower incubation temperatures, HSP70 expression was noted to be downregulated and upregulated at the embryonic ages of 12–14 and 15–17 days, respectively [115]. On a similar note, Leandro et al. [43] observed an increase in HSP70 synthesis in the breast muscles and the heart of CS embryos.

## 4. Feed Restriction (FR)

Feed restriction is a routine practice in husbandry to control obesity among meat-type chickens used for breeding stock. Broiler chickens are feed-deprived for 8 to 10 h before catching and transporting to reduce the risk of contamination of carcasses [116]. It is known that the demand for feed is inflexible in most animals, and its restriction could be stressful [117]. Controlling feed intake by limiting the duration, amount, or time of feeding may elevate circulating concentrations of CORT and heterophil to lymphocyte ratios (HLR) in poultry [47,48,49,118]. Zulkifli et al. [44] were the first to demonstrate the effects of FR on the expression of HSP70 in chickens. The authors subjected chicks to 40% and 60% feed restriction at 4, 5, and 6 days of age and noted a significant increase in HSP70 expression in the brain. Other researchers also noted the elevated HSP70 expression following FR in chickens [46,49,87,105,119]. Najafi et al. [47] subjected broiler breeder chickens to 75%, 60%, 45%, and 30% of ad libitum feed intake and reported that, although FR considerably increased brain HSP 70 levels, the authors found no evidence of a linear or curvilinear link between brain HSP 70 and feed restriction levels. Broiler chicks were given 40% and 60% feed restrictions at 4, 5, and 6 days of age, and Zulkifli et al. [44] found that both groups had comparable levels of brain HSP 70. Therefore, it appears that the magnitude of HSP expression may not accurately reflect the severity of feed limitation.

Work on avian and mammalian species has shown that neonatal stresses, while many of the animals’ systems are still developing, can elicit long-lasting changes in the physiological response to environmental challenges [120,121]. Early-age FR was reported to enhance thermotolerance and disease resistance in broiler chickens at market age [122,123,124]. Zulkifli et al. [44] reported that the acquired enhanced heat tolerance resulting from early-age fasting in broiler chickens was associated with improved HSP70 response. Liew et al. [105] demonstrated that combining FR and heat treatment during the neonatal stage enhanced weight gain and resistance to infectious bursal disease in heat-stressed broiler chickens by evoking a greater HSP70 response. The benefits of feed restriction associated with HSP expression are shown in Figure 2. Craig [125] suggested that neonatal stresses may have evoked HSP mRNA transcription, but the RNA may have been ‘sequestered’ and not translated, until exposure to heat challenges later in life. However, stressful events during the neonatal stage without corresponding increases in the synthesis and release of CORT may not help a bird to respond to later stressors [49,122]. This phenomenon could be associated with the profound role of circulating CORT in evoking HSP expression and concomitant stress resilience in poultry [45,46,126].

## 5. Pre-Slaughter Operations

The pre-slaughter operation, which involves feed withdrawal, catching, crating, transporting, lairaging, and shackling, is physiologically stressful and may compromise the welfare of poultry [127,128,129]. Circulating concentrations of CORT and HLR are standard physiological indices to measure the stress associated with pre-slaughter practices in poultry [50,130,131,132]. Zulkifli et al. [50] subjected low-fear and high-fear chickens to 3 h of crating and measured their serum concentrations of CORT and HLR, and their brain HSP 70 responses. The authors reported that crating elevated HLR and HSP70 expression, irrespective of the fear responder group. However, high-fear chickens had higher serum concentrations of CORT than their low-fear counterparts, following 3 h of crating. Elevated HSP70 protein and mRNA expression following road transportation has also been reported in newly hatched chicks [51]. Duncan [133], as measured by plasma concentrations of CORT, indicated that 40 min of transport was more stressful than if the crates were left stationary in the same vehicle for 40 min. On the contrary, based on measurements of HSP70 mRNA, plasma hormone, and rectal temperature, Delezie et al. [53] concluded that the influence of crate density on the well-being of transported chickens was more significant than that of pre-catching feed withdrawal and transit time.

Transporting chickens for 1 h or longer during summer increased HSP70 expression in the muscle of chickens [52]. The authors reported no evidence of meat quality degradation with transit time. They proposed that the increase in HSP70 production interacted with other proteins in cells and changed their function, protecting the cells from the damaging effects of stressors. Al-Aqil and Zulkifli [87] reported that broilers raised in naturally ventilated houses that were exposed to a greater variety of visual and auditory stimuli had enhanced brain HSP70 expression, compared to those in environmentally controlled pens following road transportation. The authors noted a negative relationship between brain HSP density and other physiological stress indices. Thus, the improved tolerance to transport stress in birds raised in open-sided houses could be associated with augmented HSP70 expression. Theoretically, an increase in the synthesis of HSP70 is critical during the perturbation of homeostasis to protect the cells and organs from the harmful effects of stressors [134].

## 6. Social Stress

Domestic birds are known for their gregarious behaviour, which often influences their feeding pattern, nest-site selection, perching, and parent–offspring interactions [135,136,137]. Social separation or sudden isolation is a potent stressor that may result in behavioural agitation and compromised welfare [138]. There is a dearth of literature on HSP responses to social stress in poultry. Hoekstra et al. [54] showed that isolation in darkness for 60 min increased myocardial HSP70 expression, but not HSP 30, 60, or 90 expression. When quail were supplemented with vitamin C and α-tocopherol and socially isolated for 120 min, their brain and heart HSP70 expression were unaffected, but the reverse was noted for the control birds [55]. Vitamin C and α-tocopherol are known to dampen stress-induced responses in poultry [139].

Interestingly, Soleimani et al. [55] showed that inhibiting adrenal steroidogenesis with metyrapone did not reduce HSP70 following social isolation. Hence, it appears that the blood levels of CORT and HSP70 induction are independent of each other. However, Zulkifli et al. [126] reported that daily administration of CORT from 14 to 20 days of age significantly augmented brain HSP70 synthesis in broilers. It can be concluded that the relationship between the hypothalamus–pituitary–adrenal axis function and the expression of HSP70 is not straightforward and may warrant further investigation.

## 7. Stocking Density

High stocking density is known to adversely affect growth performance, survivability, leg health, and foot pad dermatitis in broiler chickens [56,140,141]. Despite the detrimental effects associated with higher stocking densities, the total kg of broilers produced per unit of space increased, resulting in higher profits [142]. Hence, stocking broilers at higher densities is expected in the current commercial production. However, as measured by blood CORT and HLR, Thaxton et al. [143] concluded that stocking densities of 20–55 kg of body weight/m^2^ had a negligible influence on physiological stress in broilers. Imaeda [144] suggested that the recommended stocking density for broilers varies widely according to the housing system and climate. Najafi et al. [56] stocked broilers at 0.100 m^2^/bird or 0.063 m^2^ and exposed them to either 24 °C or 32 °C. The authors showed that, irrespective of ambient temperature, a higher stocking density increased the brain HSP70 expression of broilers. The effects of higher stocking densities on HSP70 mRNA expression in broilers have also been demonstrated by Beloor et al. [57]. However, Shewita et al. [58] indicated that dietary supplementation with vitamin C downregulated the expression of HSP70 in broilers stocked at higher densities. The effect of dietary vitamin C supplementation on HSP induction in physiologically stressed birds has also been reported [58,145,146].

## 8. Human Contact

A certain degree of human contact is inevitable in modern poultry production systems. Human interactions are usually viewed as predatory encounters, and the stress produced by the fear of the animal could directly affect farm animals’ welfare, performance, and productivity [147,148]. However, positive physical contact with human beings benefits poultry behaviour, welfare, productivity, and immunity [132,148,149,150]. Notably, the quality of stockmanship profoundly affects farm animal welfare [151,152,153]. There is the question of how positive human–animal interactions can improve productivity and modify the physiological stress response of farm animals. Al-Aqil et al. [59] subjected broiler chickens to either pleasant or unpleasant negative human handling from 1 to 28 d of age. Following 3 h of road transportation, the chickens had a lower HLR, shorter TI duration, and greater brain HSP70 expression than those who were ignored or handled unpleasantly. Thus, it can be concluded that pleasant human contact may improve stress resilience in poultry through enhanced HSP70 expression.

## 9. Heavy Metals

Heavy metal poisoning has raised widespread concern, particularly concerning animal health. Heavy metals, which are toxic and can damage birds’ organs, mainly accumulate through the food chain [154]. Heavy metals, including cadmium (Cd), lead (Pb), manganese (Mn), arsenic (As), and mercury (Hg), have been extensively studied in poultry, with a primary focus on the toxic effects and biological response of HSPs in various organs. HSPs are stress-protective molecules against oxidative stress that are triggered by heavy metal toxicity [155]. For example, Pb toxicity induced higher expression of HSP mRNA (90, 70, 60, 40, and 27) in the peripheral blood neutrophils, spleens, testes, and hearts of chickens [60,61,62,63]. In addition, both the mRNA and protein expression of HSPs (90, 70, 60, 40, and 27) were found to increase in the following two types of chicken cells: spermatogonia and Leydig cells [64]. The authors proposed that the HSPs responded to the depletion of antioxidant enzymes (SOD, GPx, CAT, and GST) caused by Pb-induced oxidative stress. As a solution, the same study demonstrated the ability of selenium (Se) supplementation to alleviate stress by reducing HSP levels in spermatogonia and Leydig cells. On a similar note, Xing et al. [60] found a reduction in the level of HSPs (HSP90, HSP70, HSP60, HSP40, and HSP27) after Se supplementation, which Pb initially induced in chicken neutrophils. The results were further confirmed by Wang et al. [65] in chicken testes, in which Se intervention attenuated Pb-induced mRNA expression of HSPs. Interestingly, different expression patterns of HSPs were observed for Cd compared to other heavy metals. Cd was found to significantly reduce the level of mRNA expression of HSFs (1, 2, and 3) in the chickens’ cerebellum compared to the control group. In addition, it was found that the mRNA and protein expression of HSPs (10, 25, 27, 40, 47, 60, 70, 90, and 110) decreased significantly with Cd exposure, explaining the possibility of cerebral damage in chickens [66]. A similar reduction at the HSP transcription level was reported by Chen et al. [67] and Zhang et al. [156] in chicken lymphocytes and peripheral blood neutrophils, respectively. The discrepancies between these studies could stem from the differences in the target organ. However, they found that dietary Se supplementation can increase the expression of HSPs, thereby exerting a protective function against Cd-stimulated toxic effects. The formulation of nano-Se was found to activate the expression of HSPs, indicating tolerance to Cd stress in chicken cerebellar tissue [66]. Moreover, Liu et al. [68] found dose- and time-dependent effects on HSPs associated with inflammatory injury and Mn toxicity in chicken livers. Similarly, Huang et al. [61] noted the time-dependent impact on HSP 90, 70, and 60 in chicken hearts after Pb exposure, and Liu et al. [157] noticed a dose-dependent effect of HSP90 and 60 in Mn-exposed chicken kidneys. As is the case with other heavy metals, the concentration of As increased Hsp27, Hsp40, Hsp60, Hsp70 and Hsp90 mRNA levels in the brain tissue, immune organs and gastrointestinal tract of chickens, compared to the control group [69,70,71,72]. Further confirmation of HSP70 and HSP60 protein expression was consistent with the mRNA expression during sub-chronic exposure to As in the brains of chickens, suggesting a neuroprotective function of HSPs against heavy metal-induced tissue damage [69].

## 10. Mycotoxin

Mycotoxins are secondary fungal metabolites that are commonly associated with the contamination of feed and raw food materials, resulting in significant impacts on human and animal health [158]. Mycotoxins of importance for the poultry industry are mainly produced by fungi of the genera *Aspergillus*, *Fusarium*, or *Penicillium* [159]. An initial study of the relationship between HSP70 expression and mycotoxins was performed using in vitro systems, such as HepG2 and Vero cells [160,161]. These studies demonstrated a dose-dependent increase in HSP70 protein expression, even at non-cytotoxic concentrations, following the addition of zearalenone (ZEN), citrinin (CTN), and T2 toxin (T2) mycotoxins. This finding is consistent with the effects of T2 toxins, increasing the HSP70 concentration in various hepatic cell culture models of chicken origin in vitro [73]. It was hypothesised that the higher expression of HSP70 in different cell lines could be caused by oxidative stress, systemic inflammation, or mycotoxin-induced tissue damage [160,162]. In poultry, broilers fed with a naturally contaminated diet (NCD) that contained zearalenone (ZEN), 281.92 mg/kg; fumonisin (FUM), 5874.38 mg/kg; aflatoxin (AFL), 102.08 mg/kg; and deoxynivalenol (DON), 2038.96 mg/kg, were found to demonstrate significantly higher liver HSP70 mRNA expression compared to the group fed the control diet [74]. However, it was found that broilers fed an NCD supplemented with a yeast cell wall absorbent (2 g/kg) showed significantly reduced HSP70 mRNA expression. A recent study by Paraskeuas et al. [75] documented the protective roles of HSPs in the gut of broilers fed a diet that contained FUM (20 mg/kg) and DON (5 mg/kg). The study demonstrated DON-induced upregulation of HSP90 expression levels in the cecum and ileum, while FUM induced an increase in HSP70 levels in the jejunum and cecum in 39-day-old broilers. In line with this, previous studies on 42-day-old broilers have shown increased levels of HSP70 in the liver [74] and spleen [162] caused by *Fusarium* mycotoxins. Earlier work on mycotoxins has focused on HSP70 alone. Hence, further studies are needed to determine the effects of various mycotoxins on other HSP chaperones.

## 11. Factors Modulating HSP Expression

### 11.1. Nutrition

There is considerable evidence that probiotic, prebiotic, and synbiotic supplementation could ameliorate the adverse effects of thermal [163,164,165,166,167] and non-thermal [168,169,170] stresses on poultry’s performance and well-being. Hence, these supplements are expected to influence the expression of HSPs in response to environmental stresses. Hosseini et al. [171] and Rokade et al. [172] showed that mannan-oligosaccharide (MOS) prebiotics alleviated some of the detrimental consequences of heat stress and hot, dry summers, respectively, in chickens by downregulating the mRNA expression of HSP70 in the liver, breast muscle, and jejunum. Similarly, Varasteh et al. [164] reported that galactooligosaccharide prebiotics reduced HSP70 mRNA expression in the jejunum and ileum of chickens exposed to high temperatures. Prebiotics may enhance beneficial gut microorganisms’ population, producing bioactive compounds that could prevent oxidative damage and ultimately lower the expression of HSP [164,172].

It has been reported that probiotic-fed birds experience the most significant benefits during stressful conditions [163,173]. Wang et al. [165] showed that the concentration and expression of HSP70 decreased in the livers of heat-stressed broilers fed with one strain of *Bacillus subtilis*. Similarly, a reduced level of HSP70 expression in the livers of laying hens was reported by Zhang et al. [174] by using a mixture of probiotics that contained *Bacillus subtilis* and *Enterococcus faecium*. Rather than using probiotics alone, several studies have combined probiotics with either other prebiotics or other types of minerals in the feed formulation. For instance, Khan et al. [175,176] tested selenium-enriched probiotics in chickens raised under high ambient temperatures. The findings showed that the expressions of HSP60, HSP70, and HSP90 were significantly downregulated in the cardiac and breast muscles of the chickens, respectively. Moreover, the profound effect of reducing the HSP expression level was better than in the control, probiotic-alone, and selenium-alone groups.

As a combination of probiotics and prebiotics, synbiotics have also been reported to improve the performance and well-being of HS birds [177]. Jiang et al. [166] noted that HSP70 levels in both the liver and hypothalamus were significantly lower in synbiotic groups compared to the control group, which concurs with the results on the effect of prebiotics and probiotics on HSP70 levels during HS [164,175]. By enhancing the birds’ adaptability to HS via the gut–immune and gut–brain axes, the reduced HSP70 levels discovered in the synbiotics-supplemented birds may be a sign of lower HS sensitivity and oxidative stress. On the contrary, Hu et al. [178] noted that HS had a negligible effect on the serum levels of HSP70 in chickens fed synbiotics.

Environmental stressors, mainly HS, can induce oxidative stress by altering the homeostasis between the antioxidant system and the generation of ROS [179]. Eventually, lipid peroxidation and oxidative damage would occur in the cellular membrane, leading to detrimental effects in poultry [180]. Therefore, vitamins C and E act as scavengers of ROS to protect the host from oxidative stress, improve immunity by modulating inflammatory signalling, maintain the proteins from alkylation damage, and significantly alleviate the harmful effects of HS in poultry [181,182,183,184]. Mahmoud et al. [145] and Jang et al. [185] noted a significant decline in the level of HSP70 expression in heat-stressed chickens supplemented with vitamins C and E, respectively. Sahin et al. [146] supplemented quail with vitamins C and E and exposed them to cyclic heat stress. The authors demonstrated that the ovary and brain HSP70 expression linearly decreased as dietary vitamin C or vitamin E supplementation increased. However, vitamin C or E supplementation did not affect HSP70 expression of the ovary or brain under thermoneutral conditions. The reduced HSP70 expression in heat-stressed birds supplemented with vitamins C and E could be associated with alleviating oxidative stress, which may modulate HSP induction [145].

The influence of dietary crude protein (CP) on poultry growth performance and carcass composition under heat stress conditions has been inconsistent [186,187,188,189,190]. In their review, Awad et al. [191] concluded that reducing dietary CP is recommended when birds are subjected to moderate, but not chronic, heat stress conditions. However, there is a dearth of information on HSP response to thermal insults in poultry fed low-CP diets. Zulkifli et al. [190] provided various levels of dietary CP (starter diet: 16.5–21.0%; finisher diet: 14.5–19.0%) to heat-stressed broilers and showed that nutrients had a negligible effect on the brain HSP70 density. The authors attributed the effect of a lack of dietary CP on HSP70 expression to inadequate amino acids in the lower-CP diets. There is a growing interest in supplementing low-CP diets with glutamine (Gln) as a practical approach to improving poultry growth performance and well-being in a stressful environment [192,193,194]. Glutamine, a conditionally essential amino acid, is known to protect and enhance the digestive structure and improve the survivability and performance of poultry during stressful conditions [195,196,197,198]. Dietary Gln supplementation increased HSP70 expression in chickens [199,200,201], and a similar expression pattern helped the broilers to cope with transport stress [202]. Glutamic acid (Glu), converted from Gln in the small intestine, maintains the intestinal physiology and metabolism [203,204]. Olubodun et al. [205] reported that combining Gln and Glu in the diet of broiler chickens improved their performance and survival rate under hot and humid tropical conditions. The authors concluded that the beneficial effect of Gln + Glu supplementation on heat-stressed chickens could be associated with enhanced HSP70 synthesis. Supplementing diets with Gln + Glu was also helpful in augmenting duodenal HSP70 synthesis and reducing the mortality rate in chicks subjected to delayed placement for 24 h. The metabolism of Gln in the hexosamine biosynthetic route was proposed to be a mechanism by which Gln + Glu supplementation augmented HSP 70 expression. O-glycosylation, nuclear translocation, and transcriptional activation of Sp1 and HSF-1 all appeared to play a role in this action [206].

### 11.2. Phytochemicals

The use of dietary phytochemicals as an alternative to antibiotics is gaining attention, due to their potential application in animal nutrition. Flavonoids, the largest group of natural antioxidants with multifunctional biological activities, are secondary metabolites derived from plants and are abundant in many foods [207,208]. Quercetin (QE) is a naturally occurring flavonoid that is widely available in various fruits and vegetables. QE properties, including its anti-inflammatory, antioxidative, antimicrobial, antiviral, antihypertensive, and antitumour properties, have been well characterised previously [209]. QE is also known to modulate heat shock transcription factor activity, and thus can inhibit HSP70 expression in vitro and in vivo [210]. Soleimani et al. showed that a quercetin-treated diet significantly reduced quails’ ability to express HSP70 compared to the controls following heat exposure [118]. A recent study by Sugito et al. [211] showed that *Salix* plants that contained a higher amount of QE compounds downregulated HSP70 expression in chicken renal tubules exposed to HS.

Interestingly, the *Salix* plant did not affect HSP70 expression in chickens that were not exposed to HS. Hence, the authors concluded that QE activity is affected by temperature and the type of stressor [212]. On a similar note, QE has been shown to alleviate the effects of cadmium-induced brain necroptosis by reducing the expression of HSPs (HSP27, HSP40, HSP60, HSP70, and HSP90), which were significantly increased before QE treatment [213]. In addition, other fruit-derived flavonoids are being actively tested for their free radical scavenging effects in poultry. For example, grape seed extract supplementation reduced HSP70 gene expression in the hearts and livers of broilers suffering from chronic HS, consistent with other research that involved genistein (a soy flavonoid) and hesperidin (a citrus flavonoid) from breast muscle samples [214,215]. A recent study by Sun et al. showed that astilbine (flavonoid) supplementation can regulate the expression of HSPs (27, 70, and 90), which were initially upregulated due to heavy metal-induced adipose tissue damage in chickens [216].

Resveratrol (trans-3,5,4′-trihydroxystilbene) is a non-flavonoid polysaccharide with numerous biological effects, including antioxidant and anti-inflammatory effects in poultry. An early study that investigated the activity of resveratrol in poultry was conducted by Sahin et al. [217] using quail. The result showed a 40% reduction in HSP70 protein levels in the liver in response to increasing dietary intake supplementation with resveratrol. It should be noted that no stress-inducing practices were introduced in the experiment, and the birds were housed in TN conditions. Under the HS conditions, Sahin et al. [218] noticed lower HSP70 and HSP90 expressions in the livers of heat-stressed quail fed resveratrol, compared to birds fed a basal diet. In addition, the beneficial effects of dietary supplementation with resveratrol (200, 400, or 600 mg/kg diet) for alleviating HS were investigated, and the results showed the decreased expression of HSP70 and HSP90 levels in the bursa, spleen, and jejunal mucosa of 42-day-old black bone chickens [219,220]. However, resveratrol was found to upregulate HSP27 and HSP90 mRNA expression in the thymus; therefore, the HSP mRNA expression pattern could be tissue- or organ-specific depending on the different resveratrol levels [219]. These results point to the role of resveratrol in combating HS and protecting chickens’ immune functions.

Curcumin, a major component of turmeric, is a yellow polyphenol derived from the zingiber plant and is easily ingested by animals [221]. Several studies have shown the beneficial effects of curcumin in terms of improving poultry growth performance under HS [222,223,224]. The mechanism of action of supplemental curcumin on HSP expression in poultry was investigated by Sahin et al. [225]. It was found that the HSP70 levels in hepatic cells were higher in heat-stressed quail compared to birds kept under TN conditions. However, the HSP70 levels were reported to be reduced by 26.5% with increasing curcumin supplementation. Similarly, in chickens, Zhang et al. [223] observed a rapid increase in HSP70 mRNA expression in birds that were chronically heat-stressed, while mRNA expression was significantly suppressed after curcumin administration.

In contrast, Hashemi et al. [226] reported higher HSP70 expression in heat-stressed chickens supplemented with 2% each of *Zingiber officinale* and *Zingiber zerumbet,* compared to chickens fed a basal diet alone. The discrepancies in HSP expression could be due to the percentage of curcumin introduced in the diet and the different temperature settings used during the HS challenge. Despite the discrepancies, these results clearly defined the role of curcumin as an antioxidant agent that restores mitochondrial dysfunction and suppresses the enzymes involved in ROS formation [224,226].

### 11.3. Genetics

Poultry breed and strain differences in resistance and tolerance to thermal [37,227,228,229,230] and non-thermal [231,232,233,234] stressors have been well documented. Soleimani et al. [37] compared the response of red junglefowl (RJF), village fowl (VF), and commercial broilers (CB) to acute heat stress at a common chronological age and a common body weight. In both comparisons, the RJF showed a lower HLR, higher plasma CORT concentrations, and higher HSP 70 expressions than the VF and CB. The authors concluded that domestication and selective breeding resulted in the individuals being more susceptible to stress rather than resistant. Similarly, Cedraz et al. [235] demonstrated the higher expression of HSP70 and HSP90 in local chicken breeds compared to commercial broiler lines during HS. The authors reported that the local breeds appeared to be calm, while the commercial broilers showed more significant discomfort in response to thermal stress. It was reported that the local Egyptian chicken strains expressed a higher level of HSP70 and showed a better ability to withstand HS than the commercial strains [236,237].

Work on the genetic differences in HSP reactions to non-thermal stress in poultry is limited. Sohn et al. [238] reported the higher expression of HSP70 and HSP90-α among single-comb White Leghorns compared to native Korean chicken breeds in response to a higher stocking density. Thus, the White Leghorns appeared to be more susceptible to overcrowding stress than the Korean native chickens.

### 11.4. Age

Work on human beings and laboratory animals has shown that ageing is often accompanied by neurodegeneration and modified HPA axis function in response to psychological and physical stressors [239,240]. The diminished ability to adapt to stress due to ageing has been associated with impaired HSP response. In their review, Hu et al. [241] indicated that the ability of ageing rat hearts to synthesise HSP27 in response to stress was dramatically reduced. Wang et al. [242] hypothesised that heat shock factor (HSF) 1 activity decreases as cells and organs age, depending on the cell and organ types. Additionally, HSF1 degrades, instead of activates, when ageing cells are exposed to pressure challenges, exacerbating the problem, and ultimately resulting in cell death. Working with 21-day-old and 270-day-old Japanese quail, Soleimani et al. [46] concluded that ageing, due to lower glucocorticoid receptor levels, weakened the capacity of birds to terminate CORT secretion and induce HSP 70 expression. Similarly, Lowman et al. [243] showed that younger chickens (3 weeks old) showed higher expression of HSP90, HSP60 and HSP70 mRNA than their older counterparts. The authors hypothesised that, being less developed, the three-week-old birds might have less cellular stress memory than the older birds.

### 11.5. Gender

Figueiredo et al. [244] determined the possible effects of gender on the ability to express HSP and showed that males had greater levels of HSP70 mRNA than female chickens. These findings concur with those of Lowman et al. [243], who showed that male broilers had significantly higher expression levels of HSP90A, HSP90AA, and HSP90β than females. Wang and Edens [245] reported the relationship between testosterone levels and HSP expression in chickens. Paroo et al. [246] suggested that the mechanism by which estrogen influenced HSP response was through non-genomic membrane stabilisation. Paroo et al. [247] showed that estradiol treatment attenuated post-exercise HSP 72 expressions in rats’ hearts, liver, lungs, and red and white vastus muscles.

### 11.6. Fear

According to Jones [248], fear is “an emotional (psychophysiological) response to perceived danger”. Fear and stress are not synonymous, but fear is a potent stressor, especially if the frightening stimulation is severe, extended or unavoidable [249,250]. Tonic immobility (TI) is an established indicator of fear in poultry [148,251]. Zulkifli et al. [50] studied the relationship between TI and HSP expression in broilers subjected to heat stress or crating. As measured by TI duration, the authors classified broiler chickens as low-fear (LF) or high-fear (HF) responders. Following three hours of heat exposure, the brain HSP70 density was greater for the HF group than the LF group. However, the LF and HF groups showed similar HSP 70 expressions after crating. The relationship between TI and HSP70 induction was also reported by Al-Aqil et al. [59]. Following three hours of road transportation, the authors noted that the chickens with enhanced HSP70 expression showed a reduced TI duration and physiological stress response. It could be concluded that the activation of HSP 70 appears to be related to the physiological stress reactivity of birds, with different degree of underlying fearfulness [50].

## 12. Conclusions

The intensive poultry production system emphasises enhancing productivity and reducing costs to remain economically viable. However, productivity improvements might result in trade-offs with other concerns, such as animal well-being. Poultry employ behavioural and physiological stress reactions to cope with the demanding intensive production environment to maintain homeostasis. When chickens are exposed to thermal and non-thermal stressors, HSPs are upregulated, as is the case in other living organisms. Most HSPs serve as molecular chaperones, assisting organisms in overcoming internal and external stresses. Changes in the expression of HSP following noxious stimuli suggested that these proteins can be used to gauge physiological stress in avian species and are, thus, another biomarker for well-being. Because birds exhibit a wide range of reactions to environmental stressors, understanding the cellular and molecular mechanisms of the stress response is necessary. Detailed research that defines the mechanisms of stress resilience and understanding the nature of heat shock signalling proteins and gene expression will be valuable for developing operational strategies to alleviate the detrimental consequences of environmental stresses in poultry. The potential cost–benefit trade-off of HSP overexpression in poultry also merits further investigation.

## Figures and Tables

**Figure 1 animals-13-00317-f001:**
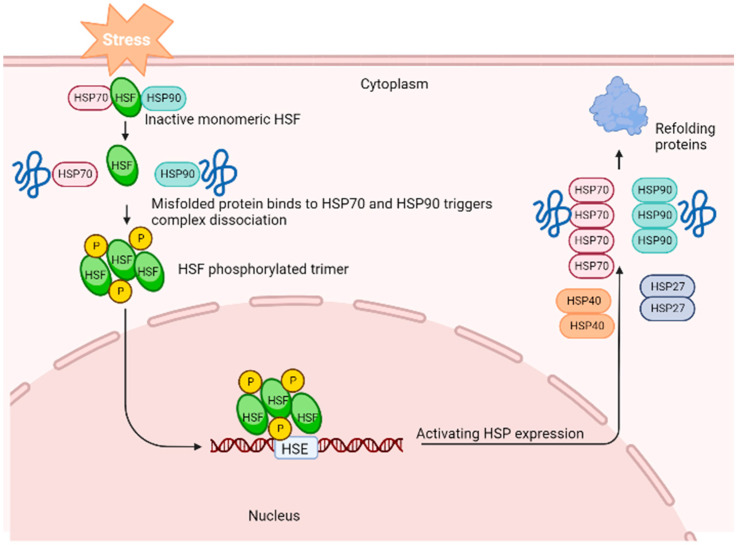
Activation of heat shock proteins’ activity in response to stressors. (Created with BioRender.com).

**Figure 2 animals-13-00317-f002:**
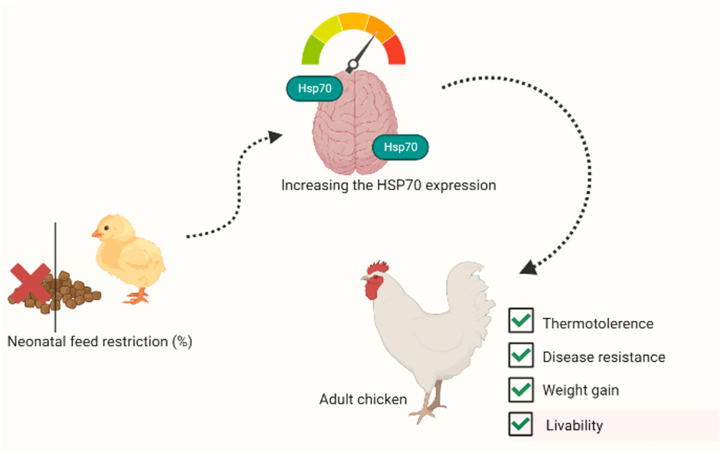
The benefits of feed restriction associated with HSP expression. (Created with BioRender.com).

**Table 1 animals-13-00317-t001:** The expression pattern of heat shock proteins associated with different major stressors.

Stress	Types of Samples	Major Classes of HSP	Type of Expression	Effects	Reference
Thermal stressors					
Heat stress					
	Leukocytes, testes; bursa of Fabricius	70	mRNA	HSP70: ↑	[24]
	Heart; lungs	90, 70, 29; 27	Protein	HSP 90, 70, 29; 27: ↑	[25]
	Heart	90, 70; 60	mRNA	2 h of heat stress; HSP 60, 70; 90: ↑ 3 h of heat stress; HSP 60; 90; ↓, HSP 70: -	[26]
			Protein	2 h of heat stress; HSP 60, 70; 90: ↑ 3 h of heat stress; HSP 90: ↓, HSP 60; 70: -	
	Muscle	90; 70	mRNA	HSP 70: ↑, HSP 90: -	[27]
	Liver			HSP 70: ↑, HSP 90: -	
	Heart			HSP 90: ↑, HSP 70: -	
	Liver	90	mRNA	2 h of heat stress; HSP 90: ↑ 10 h of heat stress; HSP 90: -	[28]
			Protein	2 h of heat stress; HSP 90: ↑ 10 h of heat stress; HSP 90: -	
	Kidney		mRNA	2 h of heat stress; HSP 90: ↑ 10 h of heat stress; HSP 90: -	
			Protein	2 h of heat stress; HSP 90: ↑ 10 h of heat stress; HSP 90: ↑	
	Heart		mRNA	2 h of heat stress; HSP 90: ↑ 10 h of heat stress; HSP 90: ↓	
			Protein	2 h of heat stress; HSP 90: ↑ 10 h of heat stress; HSP 90: ↑	
	Testes	90AA1, 70; 25	mRNA	HSP 90AA1, 70; 25: ↑	[29]
	Testes	90AA1, 70, 25, A8, A5, H1; D1	mRNA	HSP 90AA1, 70, 25, A8, A5, H1; D1: ↑	[30]
	Blood	90, 70, 60; 27	mRNA	HSP 90, 70, 60; 27: ↓	[31]
	Liver	70, 60; 40	mRNA	HSP 70, 60; 40: ↑	[32]
		70, 60; 40	Protein	HSP 40: ↑ HSP 70; 60: -	
	Brain	70	mRNA	HSP 70: ↑	[33]
	Duodenum	70, 60; 47	mRNA	3 h of heat stress; HSP 70, 60; 47: ↑	
			Protein	6 h of heat stress; HSP 70; 60: ↑ 3 h of heat stress; HSP 47: ↑	
	Jejunum		mRNA	3 h of heat stress; HSP 70, 60; 47: ↑	[34]
			Protein	6 h of heat stress; HSP 70; 60: ↑	
	Ileum		mRNA	3 h of heat stress; HSP 70, 60; 47: ↑	
			Protein	3 h of heat stress: HSP 47: ↑	
	Muscle	90; 70	mRNA	HSP 90; 70: -	[27]
	Liver			HSP 90: ↑, HSP 70: -	
	Heart			HSP 70: ↑, HSP 90: -	
	Kidney, brain; liver	70	Protein	HSP 70: ↓	[35]
	Muscle, heart; brain	90; 60	mRNA	D10; HSP 90; 60: ↑ D28; HSP 90; 60: ↑	[36]
	Liver	90β, 70; 27	mRNA; protein	HSP 90β, 70; 27: ↓	[37]
Cold stress					
	Heart	70, 60, 40, 27	mRNA	HSP 70, 60, 40; 27: ↑	[38]
		90	mRNA; protein	HSP 90: ↓	
	Immune system organs	90, 70, 60, 40; 27	Protein	HSP 90, 70, 60, 40; 27:	[39]
	Heart	90, 70; 27	mRNA	HSP 90, 70; 27: ↓	[40]
	Ileum	90, 70; 60	Protein	HSP 90, 70; 60: ↓	[41]
	Hepatic cells	70	mRNA	HSP 70: ↓	[42]
	Splenic cells	70	mRNA	HSP 70: ↓	
	Heart; muscle	70	Protein	HSP 70: ↑	[43]
Feed restriction					
	Brain	70	Protein	HSP 70: ↑	[44]
	Brain	70	Protein	HSP 70: ↑	[45]
	Hippocampus	70	mRNA	HSP 70: ↑	[46]
	Brain	70	Protein	HSP 70: ↑	[47]
	Brain	70	Protein	HSP 70: ↑	[48]
	Brain	70	Protein	HSP 70: ↑	[49]
Pre-slaughter operations					
Crating	Brain	70	Protein	HSP 70: ↑	[50]
Transportation	Heart	110, 90, 70, 60, 47; 40	mRNA	2 h of transportation stress; HSP 110, 90, 70, 60, 47; 40: ↑ 8 h of transportation stress; HSP 110, 90, 70, 60, 47; 40: ↓	[51]
		90, 70; 60	Protein	2 h of transportation stress; HSP 90, 70; 60: ↑ 8 h of transportation stress; HSP 90, 70; 60: ↓	
	Muscle	70	mRNA; protein	HSP 70: ↑	[52]
Crating and transportation	Liver	70	mRNA	HSP 70: ↑	[53]
Social stress					
	Heart	90, 70, 60; 30	Protein	HSP 70: ↑ HSP 90, 60; 30: -	[54]
	Heart; brain	70	Protein	HSP 70: ↑	[55]
Stocking density					
	Brain	70	Protein	HSP 70: ↑	[56]
	Liver	90; 70	mRNA	HSP 70: ↑ HSP 90: -	[57]
	Heart; liver	90; 70	mRNA	HSP 90; 70: ↑	[58]
Human contact					
	Brain	70	Protein	HSP 70: ↑	[59]
Heavy metal					
	Neutrophils, spleen, testes; heart	90, 70, 60, 40; 27	mRNA	HSP 90, 70, 60, 40; 27: ↑	[60,61,62,63]
	Spermatogonia; testis	90, 70, 60, 40; 27	mRNA; protein	HSP 90, 70, 60, 40; 27: ↑	[64]
	Testes	90, 70; 60	Protein	HSP 90, 70; 60: ↑	[65]
	Cerebellum	110, 90, 70, 60, 47, 40, 27, 25; 10	mRNA; protein	HSP 110, 90, 70, 60, 47, 40, 27, 25; 10: ↓	[66]
	Lymphocytes	90, 70, 60, 40; 27	mRNA	HSP 90, 70, 60, 40; 27: ↓	[67]
	Liver	90, 70, 60, 40; 27	mRNA	HSP 90, 70, 60, 40; 27: ↑	[68]
	Brain, spleen, thymus, jejunum, ileum; duodenum	90, 70, 60, 40; 27	mRNA	HSP 90, 70, 60, 40; 27: ↑	[69,70,71,72]
Mycotoxins					
	Liver	70	Protein	HSP 70: ↑	[73]
	Liver	70	mRNA	HSP 70: ↑	[74]
	Cecum, ileum; jejunum	90, 70	mRNA	HSP 90: (except jejunum) ↑	[75]
				HSP 70: (except ileum) ↑	

Note: HSP: heat shock protein, h: hour, ↑: increases significantly compared to control, ↓ decreases significantly compared to control; -: no significant changes compared to control.

## Data Availability

The supporting findings of this study are available within the article.

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
