# Peer review of "Heat Shock Protein Response to Stress in Poultry: A Review"

_animals, 2023, doi:10.3390/ani13020317_

Round 1

Reviewer 1 Report (Previous Reviewer 1)

I have only to comments. See the text. 

Author Response

Reviewer 2 Report (Previous Reviewer 2)

No more suggestions for changes in the manuscript. I belive it was substantially improved. 

Author Response

Reviewer 2 is satisfied with the corrections round 1. No any further corrections are asked for in round 2.

This manuscript is a resubmission of an earlier submission. The following is a list of the peer review reports and author responses from that submission.

Round 1

Reviewer 1 Report

All comments and suggestins are introduced in to the text. 

Reviewer 2 Report

Title: Heat Shock Protein Response to Stress in Poultry: A review

General comments

The tittle clearly describes this review on heat shock protein response to several classes of stressors, with emphasis on heat stress in poultry. Some of the authors in this group have previously published papers on related subjects. Recently, competing reviews have been published on similar or related topics, some of them in “Animals” (for instance Shehata et. al. 2020, https://doi.org/10.3390/ani10122407; Perini et al., 2021, https://doi.org/10.3390/ani11010046; Goel et al. 2021, https://doi.org/10.1186/s40104-020-00523-5). In contrast with them, the present review does not present any Tables, Figures or Diagrams, but only plain text, which makes it less attractive to readers. It will need an English language review, due to some grammar issues and inadequate vocabulary.

Specific comments

Abstract

Lines 21-22: move “to maintain Homeostasis” to the end of the sentence, after “(HSPs)”.

Introduction

Line 42: change “disease organisms” to “pathogens”.

Line 54: there is no need to cite the authors’ names.

Line 58: there is no need to cite the authors’ names.

Line 63: use “proteins”, not “protein”.

Line 66: use “acquisition of stress tolerance” not “acquire stress tolerance”.

Line 69: use “these proteins”, not “the protein”.

Line 74: delete “organismal”.

Lines 79-80: use “chromosomal puffs”, not “chromosome stuff”.

Line 106: change “discusses” to “to discuss”.

Heat Stress

Line 109: “a roller-coaster ride” – the use of colloquial language is not adequate in a scientific paper. Please rewrite.

Line 115: use “increase” (not “increases”).

Line 118: change to “HSPs expression”.

Line 150: “the concept of HSPs being time and tissue-dependent”

Line 159: “can be explained by mRNA turnover”.

Lines 167-168: the sentence: “a pertinent role of the muscle is the least and the heart the most responsive towards HS following HSPs expression similar to other avian studies [41,52-168 55].” is unclear. What do you mean? Please rephrase.

Line 184: Change to “The promoter of the HSP70 gene”.

Lines 184-185: you said, “the promoter was affected by methylation”. This is vague. Be specific. How affected? Was it hypermethylated or hypomethylated?

You said, “which altered protein expression and response to HS during post-hatch life.” Again, “altered” is vague. Has protein expression increased or decreased? Be specific.

Lines 193-201: the entire section describing the study by Greene et al. is confusing. In especial what do you mean by “All the HSPs (90, 70, 60 and 27) and HSF (1, 2, 3, and 192 4) were not significantly expressed during HS, and also a group of broilers fed with phytobiotics feed additive (PFA).”??? This makes no sense.

Line 200: “the authors selected blood”.

Line 202: I believe “emphasize” is not appropriate. Please use “involve or encompass”.

Line 203: I suggest starting a new paragraph at “Kang and Shim [68] investigated…”

Lines 208-211: “On the contrary, all HSP mRNA expressions of HSPs (70, 60, 40, HSF1 and HSF3) were higher than in non-heat conditioned broilers and control. In comparison, only HSP40 and HSP70 protein and mRNA expressions increased in non-heat-conditioned chickens subjected to HS”. Whose findings are these? I believe this sentence belongs to the previous paragraph. That is one of the reasons to start a new paragraph at “Kang and Shim…”

Lines 219-220: “while the other is a low level of expression [70,73,76].” This makes no sense. Please clarify.

Line 221: delete “and”.

Lines 223-225: The sentence “Nevertheless, 42 d of chronic HS chickens subjected to pre-thermal conditioning showed a significant reduction of mRNA and protein expressions of HSP90ß, 70 and 27 compared to control” makes no sense. Please rephrase to clarify.

Line 233: do you mean “are not yet fully understood”?

Line 251: use “ages”, instead of “age”.

Line 272: “have shown”

Line 384: Bacillus subtilis

Line 396: add “results” after “those”.

Line 407: add “damage” after “alkylation”

Line 412: “the ovarian and brain…”Line 423: at the endo of the line use “:”, not “;”

Line 425: “lack of effect of dietary CP..”.

Line 433: delete “profoundly”.

Line 436: “survivability rate under hot and humid tropical conditions”.

Line 444: “Genetics”, not “Genetic”.

Lines 498-499: “suggested that these proteins”.

Reviewer 3 Report

I recommend that the authors summarize the recent studies in tables.

The molecular mechanisms of HSP participation in acquisition of stress need further detailed explanation.

regulating effects of various phytochemicals on HSP need further explanation.

Other important stresses, such as heavy metals and mycotoxins, should be discussed in this paper.
